# The Importance of Providing Play and Learning Materials for Children with Physical Disabilities in Saudi Arabia: The Perceptions of Parents

**DOI:** 10.3390/ijerph19052986

**Published:** 2022-03-03

**Authors:** Mohaned G. Abed, Todd K. Shackelford

**Affiliations:** 1Department of Special Education, Faculty of Educational Graduate Studies, King Abdulaziz University, Jeddah 21589, Saudi Arabia; mabed@kau.edu.sa; 2Department of Psychology, Oakland University, Rochester, MI 48306, USA

**Keywords:** parenting, play materials, learning materials, physical disabilities, children, Saudi Arabia

## Abstract

The purposes of this exploratory, qualitative research are to (a) examine the parental role in providing materials to facilitate learning and play for children with physical disabilities, and (b) identify the difficulties expressed by parents that affect the support provided to their children with physical disabilities. We conducted semi-structured interviews with 21 Saudi Arabian parents of children with physical disabilities to identify the parental role in providing play and learning materials as well as the challenges identified by parents in providing this support. Parents recommended that they initiate and maintain contact with other parents of children with physical disabilities and with associated organisations. Parents also recommended that teachers welcome parents to be involved in support of their children with physical disabilities, especially in the provision of play and learning materials. The authors conclude that disability awareness programs for peers and staff members may improve physical and psychological health for children with physical disabilities.

## 1. Introduction

Appropriate play and learning materials are important for the successful development of children, particularly in the early years. Witt [1] notes that appropriate play opportunities encourage learning in a number of different ways and can facilitate the development of independence, self-confidence, social competence, and physical skills. For children with physical disabilities, play and learning warrant substantial support and attention. In contrast to non-disabled children, those with physical disabilities often require more attention, especially when play is used to improve holistic development. Mwathi [2,3] notes that children with physical disabilities often depend on their immediate support network and may participate only minimally within and outside of school. This may be because children with physical disabilities lack appropriate play and learning materials.

Different measures have been enacted to address the difficulties faced by people with physical disabilities. Using the Convention on Rights of Persons with Disabilities (CRPD), the United Nations facilitated a formal agreement in 2006 [4,5]. This Convention had been ratified by 96 countries four years later. The adoption of new national laws and the repeal of outdated laws is becoming more commonplace, with persons with physical disabilities recognised as having the same rights to cultural life, education, and employment as those without physical disabilities. Saudi Arabia has ratified the Convention. However, there is still much to be achieved in terms of encouraging the overall welfare of persons with physical disabilities.

### 1.1. Children with Physical Disabilities

The term “physical disabilities” in the context of this study is used in reference to a physical condition which is different from what is considered normal, with the result that a modification in school practices is necessary to facilitate a child achieving their full potential [6]. Such physical disabilities have an impact on normal, daily functioning as a result of limitations in abilities to use limbs or hands, or limitations in mobility, strength, or trunk control. Such restrictions impact overall mobility such that the person may need orthopaedic appliances that aid mobility, such as a wheelchair or crutches.

In the view of Sciarra [7], children with physical disabilities need to be viewed as normal to the greatest possible degree. This implies that teachers and parents responsible for children with physical disabilities need to focus on strengths rather than weaknesses. This, in turn, helps the child to feel successful [7].

### 1.2. Children with Physical Disabilities in Saudi Arabia

Notwithstanding the fact that state policy in Saudi Arabia is founded in and promises to respect the outcomes of the CRPD, which requires equal access to education for children with physical disabilities, several studies e.g., [8] have concluded that educational marginalisation of Saudi children with physical disabilities still remains.

Concerning the significance of the availability of sufficient facilities and equipment, Al Mouter [9] investigated the reasons students with physical disabilities dropped out of public schools in the Saudi capital of Riyadh. The study assessed 171 students with physical disabilities from a number of single-sex schools. Females constituted 53% of the aggregate participants. Al Mouter [9] used a mixed-methods approach, including school observations, interviews with parents, and surveys of students. The results showed that 112 of the students with physical disabilities dropped out of the education system because of limited access to facilities for students with physical disabilities. For instance, over 86% of the schools did not have access through wheelchair ramps. Additionally, 93% of the schools did not have special toilet facilities to serve students with physical disabilities. The results of this study triggered an investigation by the Ministry of Education in Saudi Arabia in relation to the lack of access facilities for students with physical disabilities.

An area of recent interest to Saudi educators and policy advocates is the degree to which children with physical disabilities could be accommodated in pre-schools if there were appropriate facilities. In Al Mouter’s [9] study, two pre-schools had much better facilities than others. These pre-schools had been deliberately built to meet the needs of children with physical disabilities, ensuring appropriate toilet facilities, resource rooms, and easy access. On the other hand, the remaining pre-schools lacked these resources, making it difficult to include students with physical disabilities.

### 1.3. Benefits of Learning and Play Materials for Children with Physical Disabilities

Participating in physical activity on a regular basis has a number of benefits. These benefits include developing and maintaining physical health [10,11] and psychological health [12,13]. Physical activity also often facilitates social engagement. When children with physical disabilities participate in regular activity, they also often enjoy other therapeutic benefits [14].

According to Cook, Richardson-Gibbs, Nielsen and Klein [15], children with and without physical disabilities benefit from playing. The same scholars note that the outside environment provides a change of scenery and prospects that helps facilitate independence and the development of physical and social skills. Normally, when compared to indoor environments, outside settings may require various physical abilities. For instance, it may be necessary to modify playground equipment so that children with physical disabilities can attain a maximum range of muscle control, reach, visual contact, and motion. Adjusting the outside settings for children using wheelchairs may also be achieved through placing equipment at lower levels [16].

For those whose aim is to assist children with physical disabilities, resources for learning and playing can play an important role. Such resources make it possible for these children to participate in different activities, whether learning or playing. These experiences and activities can improve holistic development, which includes physical development [17], the development of self-care skills [18], the development of social skills [19], language development [20], emotional development [21], and cognitive development [22].

### 1.4. Problem Statement

A number of studies have demonstrated that play, particularly in the early years, facilitates both healthy development and learning. Nonetheless, ensuring that those with physical disabilities have the appropriate materials for play and learning can be challenging. This means that, for parents of children with physical disabilities, there are additional responsibilities for ensuring that suitable play and learning materials are available, and that the children have appropriate assistive devices to facilitate play and learning. The present study seeks to identify the role that Saudi Arabian parents play in providing materials for playing and learning, and also to identify the disabilities parents face in the provision of such resources.

### 1.5. Significance of the Study

This study investigates the parents’ role in providing materials for use in playing and learning for children with physical disabilities. This is the first such investigation in Saudi Arabia. This research therefore may provide policymakers with information that could be useful in developing policies in support of providing necessary play and learning materials for Saudi children with physical disabilities. The findings generated from this study also may be useful for teacher training institutions in Saudi Arabia, where a greater degree of attention might be directed to materials that facilitate play and learning for children with physical disabilities—and to encourage teachers to provide suitable resources.

### 1.6. Study Questions


What role do parents play in making available materials for learning and playing to children with physical disabilities?What factors affect parents’ abilities to provide learning and play materials to children with physical challenges?


## 2. Methods

### Procedures, Participants, and Data Analysis Plan

The current study makes use of a qualitative research method: semi-structured interviews. Such an approach may be described as an in-depth and extensive investigation of a specific situation. In this research, a qualitative design—specifically, semi-structured interviews—was identified as especially valuable in facilitating focus on the views of parents of children with physical disabilities [23]. In addition to allowing for extended discussions and making it possible for participants to provide details in relation to their personal opinions and experiences, this methodology affords the benefits of structured interviews, including focused questions presented in a systematic manner [24]. The semi-structured interview technique employed in the current study has been effective in previous research on early childhood education e.g., [25,26].

The participants were a convenience sample of 21 parents of children aged six to eight years old, recognised by medical and educational authorities as having physical disabilities. Table 1 presents demographic information for participating parents. The physical disabilities of children included arthritis, muscular dystrophy, arthrogryposis, cerebral palsy malformation, and spina bifida. No additional demographic information was secured about the children, to protect their identities.

We devised a semi-structured interview schedule to secure information from parents about how their children with physical disabilities are supported, as well as to identify the challenges facing parents in providing play and learning materials for their children. The interview was guided by three questions: “To what extent do you think play and learning materials are important for your child?”, “What do you think is the role of the parents of children with physical disabilities in providing play and learning materials?” and “What factors do you believe affect parents of children with physical disabilities in providing play and learning materials?” These questions were intended to address the research questions as well as previous related research [27,28,29] and from informal discussions of the senior author over many years with parents of children with physical disabilities. Parents were interviewed at a time and place of their choosing.

The senior author, a trained interviewer and educator, conducted interviews during the 2019 academic year. Interviews took approximately 50 min. The researcher recorded parents’ responses verbatim, and occasionally asked follow-up questions, for clarification. The study and its purpose were explained to all participants.

Institutional research approval was obtained from King Abdulaziz University. On behalf of the researchers, King Abdulaziz University then sent a letter to the local public school educational authority in Jeddah, Saudi Arabia to secure approval to conduct the research. Approval to conduct the research was secured from the educational authority. The letter was then transmitted to the head teachers in the schools. Informed consent was obtained from participating parents prior to their participation. To ensure that participation in the research did not have negative consequences for parents or their children, strict confidentiality was observed.

After recording, all interviews were transcribed. In line with Li and Liu [30], the software programme NVivo was used to organise the answers to each interview question to identify themes that emerged from the responses [31]. This was followed by coding of responses according to the themes identified. As reported below, three themes were identified.

For the present study, the data were collected using qualitative methods. Before collecting the data and establishing the coding process, the researcher started with a coding scheme based on the research questions. Once the codes had been established, they were developed into themes. The illustrative responses are quoted in the following sections to demonstrate the tone and narrative quality of parental perceptions.

## 3. Results

Parents’ responses were subjected to qualitative analysis. For each area addressed by the interview questions, parents’ responses were organised by the researchers into categories and themes. The following sections address parent responses in each of the areas addressed by the interview questions, in turn. We present direct quotes from parents to illustrate common responses.

### 3.1. Importance of Provision of Play and Learning Materials

Parents unanimously agreed on the importance of the provision of play and learning materials for all children, but especially for children with physical disabilities. As is the case with other children, children with physical disabilities enjoy, for example, parks and playgrounds, at school or elsewhere. However, in the absence of appropriate equipment, they may not be able to participate and, according to their parents, this may negatively affect them psychologically, including causing them to feel deficient because of their physical disabilities. One father (p7) stated that: “I make sure to provide my son with play materials and take him to playing places and consider that this is part of his treatment and rehabilitation, as it is useful when combined with physical and occupational therapy and it also helps to cope with the loss of certain capabilities and contributes to his integration in sports, social and artistic activities.”

However, there are obstacles faced by parents of children with physical disabilities, as one father (p10) mentioned: “When I go to the entertainment city near my house, the supervisor refuses to let my child enter, fearing for him to get injured and his inability to play, which forces me not to accompany him so that he does not feel deficient.” In addition, all the parents mentioned the importance of the playgrounds being modified so their children can play freely within their abilities. One father (p2) noted: “It is very important that all obstacles are removed from the place.”

Four parents (19%) reported sensing shame in the community because of the presence of their child with physical disabilities. Nonetheless, 17 participants (81%) recognised the benefits of playing for all children, including their children with physical disabilities. Although they recognised the value of formal occupational and rehabilitation therapy for their children, parents also reported valuing play for their children. As one parent (p16) stated, “Playing differs as it helps the child with disabilities to cope with the loss of certain capabilities, which are considered to be key to engaging in sports, social, and artistic activities”. Another parent (p8) added that play “develops the interests of the child living with disabilities in other activities that suits their current capabilities and helps them to integrate socially and improve their health.” A third parent (p20) emphasised that “children with special needs require special attention in everything. Hence, the most important elements that should be focused on are the activities that they practice,” and this includes play, according to this parent.

Eighteen parents (86%) commented that parents of children with physical disabilities should be educated about the ability of their children to engage in various types of play without unnecessary caution. For instance, as one mother (p3) explained: “The sense of fear among parents is passed on to children, which causes them to be reluctant to play and creates a separation wall between the child’s well-being and security, which makes him avoid the recreational side permanently.”

### 3.2. Parental Provision of Materials to Facilitate Learning and Play

All parents noted the importance of parental involvement with and encouragement of children with physical disabilities to use their abilities to the fullest extent. One mother (p9) commented that, “If he is unable to use his hands, the parents must provide the toys that can be engaged with the feet; if he is unable to use both hands and feet, for example, the parents must adjust and provide games that he can play with his face.”

Nineteen of the parents (90%) commented on the importance of a partnership with teachers and involvement of the parents in the school. One father (p4) mentioned: “The parents should request that [appropriate play and learning materials for children with physical disabilities] are available at school, and if the teacher does not provide these, parents should request them from the school principal.” Twenty parents (95%) noted that schools are not equipped or prepared to accommodate children with physical disabilities, and parents therefore have difficulties identifying a school for their children.

More than two-thirds (16; 76%) of the parents indicated that their child did not have access to suitable mobility and assistive devices at school, while the remainder of the parents (5; 24%) indicated satisfaction with the equipment and materials available both in the school and at home. Eight parents (38%) emphasised the importance of parents advocating for schools and communities to follow government requirements to provide appropriate equipment and materials for their physically challenged children. One of the mothers (p11) stated: “There is a requirement to establish playgrounds that can be accessed by children with special needs, and government municipalities should not establish playgrounds that do not meet these requirements.” One father (p13) added that “sports clubs must open their doors to children with physical disabilities and accept their registration and membership, as this a legal requirement.”

### 3.3. Factors Affecting the Provision of Play and Learning Materials

Parents identified several factors hindering provision of play and learning materials, including: An absence of or inattention to relevant laws to provide these materials (9 parents; 43%); the lack of public and private counselling centres that might help a family accept the presence of a child with physical disabilities (17 parents; 81%); social difficulties—i.e., the family in which there is a child with special needs faces a number of social problems, including negative valuation by other family members and the community (13 parents; 62%); economic difficulties—i.e., the high costs paid for education, training sessions, medical treatment, surgery, or equipment for the child with physical disabilities (20 parents; 95%); lack of playgrounds suitable for children with special needs (19 parents; 90%); and, finally, not knowing what services, equipment, or materials are available for their child with physical disabilities through the school, community, or government (17 parents; 81%).

## 4. Discussion

Children who exercise and play regularly enjoy better developmental outcomes [32]. If children with physical disabilities are to play and learn, they need to have access to appropriate learning and play materials. The current research suggests that children with physical disabilities in Saudi Arabia often do not have appropriate access to these materials. This assessment is corroborated by other studies which show that children with physical disabilities are involved in less physical activity than children without physical disabilities e.g., [33].

The majority of schools serving children with special needs in Saudi Arabia have not been adequately modified to meet the requirements of learners with physical disabilities [34]. These children and their parents report that they need not only the appropriate equipment but also medical facilities that will assist them on a daily basis, facilitate mobility, and adjust to their life [34]. These needs are acknowledged by other scholars, including Mwathi [2,3], who noted that children living with physical disabilities often must depend on others even after they leave formal education. This indicates that such individuals may not have appropriate access to programs, materials, or equipment that prepares them to live independent lives [35].

As emphasised by parents in this study, opportunities for children with physical disabilities to play outdoors are needed. Outdoor play provides an environment which is different from that of the classroom, facilitating unique opportunities for children to develop social skills, gross motor skills, and self-help skills [15]. In modifying the environment, the inaugural step is to gain a clear understanding of the character and extent of the disabilities of children. Following this, the environment can be modified to facilitate the participation of each child so that their abilities rather than their disabilities are emphasised [36]. The degree to which a child with physical disabilities feels positive about learning and play activities is linked to the degree to which his or her parents accept such a child and encourage independence [37].

Many parents of children with physical disabilities report that they feel unwelcome or uncomfortable in their children’s schools or play places. This implies that teachers and others have not fully welcomed parents to be actively involved in support of their children, especially in the provision of play and learning materials. Parents of children with physical disabilities need to demonstrate commitment to the school providing a suitable environment and appropriate materials for their children. According to Epstein [38], the rationale for cooperation between teachers and parents should be to facilitate the child’s success, not only in school but throughout life. Collaboration with the community can be achieved by ensuring that community services designed for children are accessible to children with physical disabilities.

On the basis of parents’ responses in the current research, it can be concluded that they believe that social barriers to participation (including staff, parent, and peer attitudes) had more negative effects than other kinds of barriers. These social barriers include social stereotypes about children with physical disabilities and the refusal by peers to accept children with physical disabilities [39]. These attitudes make children with physical disabilities less willing to participate [40]. While changing attitudes is a challenge, contact theory advances the view that knowing someone with a physical challenge through such contexts as working with them could have a positive impact on changing attitudes [41].

Other methods that could reduce confusion regarding what people with physical disabilities need includes conducting disability awareness programs for peers and staff members [42]. The advantage of such programs is that they can help in developing skills and knowledge about how activities can be modified [43], a situation that could encourage interaction between peers to establish an environment that is welcoming. Opportunities for children with physical disabilities to interact with children without physical disabilities do not often occur automatically [44]. Hence, such activities need to be deliberately planned so that they can deliver opportunities for children with physical disabilities to develop friendships with non-challenged peers [45] and to become aware of each other’s needs [46].

## Figures and Tables

**Table 1 ijerph-19-02986-t001:** Demographic information of parents (*n* = 21).

Variable		Frequency
Gender	Male	13
	Female	8
Age (years)	(20–30)	0
	(30–40)	9
	(40–50)	10
	(50–60)	2
Qualifications	Secondary	1
	Bachelor	19
	Master	1
Child’s age (years)	6	14
	7	5
	8	2

## Data Availability

Due to the nature of this research, participants did not agree for their identified data to be shared publicly.

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
