# Peer review of "The Importance of Providing Play and Learning Materials for Children with Physical Disabilities in Saudi Arabia: The Perceptions of Parents"

_ijerph, 2022, doi:10.3390/ijerph19052986_

Round 1

Reviewer 1 Report

Title: The Importance of Providing Play and Learning Materials for Children with Physical Disabilities in Saudi Arabia: The Perceptions of Parents

Article Type: original scientific paper

Summary

This study has investigated the role of parents in providing materials to facilitate learning and play for children with physical disabilities and also identifying the difficulties that affect the support provided to these children. This study was an interviews study. The results indicated that disability awareness programs for peers and staff members may improve physical and psychological health for children with physical disabilities.

Evaluation

Although the subject of the study was interesting to me, it seems that the correct scientific methods have not been well used to achieve the purpose of the study, so I think the current manuscript does not appear to be eligible for publication in IJERPH.  For example; 1- The abstract needs to be reviewed, the research method used, the information about the participants in the research, the methods used in the research have not been discussed. 2- The introduction is not well written and the main research question is hidden.3- Considering the different types of disabilities in children and their different needs, it is necessary to talk about the types of disabilities in children in the present study.4- Discussion should be written based on the main purpose of the research.

Author Response

Please see attached, and thank you for your time and attention.

Author Response

(The authors gave the same response as above.)

Reviewer 3 Report

This paper reports a practical and sensitive problem about the importance of playing and learning materials to children with physical disabilities. They conducted the study in Saudi Arabia from 21 family having children with physical disabilities. From the interviews from parents and the analysis, the authors summarized the factors affecting the provision of play and learning materials. More attentions, care, understanding, measures should be taken by governments, communities, schools as well as families to encourage and promote the participation of children with physical disabilities to play indoors and outdoors. Public programs such as disability awareness programs are also important. In my opinion, this manuscript can be accepted for publication after a minor revision.

Minor suggestion:

  • Line 205, child should be revised as children;
  • Similar studies about the concerns of children with physical disabilities should be introduced and discussed in this paper. For example, a similar study relating to the children with physical disabilities, see [The Study of Spatial Safety and Social Psychological Health Features of Deaf Children and Children with an Intellectual Disability in the Public School Environment Based on the Visual Access and Exposure (VAE) Model. Int. J. Environ. Res. Public Health 2021, 18, 4322. https://doi.org/10.3390/ijerph18084322].

Author Response

(The authors gave the same response as above.)

Round 2

Reviewer 1 Report

Thank for the author's revision, I think the manuscript now can be published in the current version. So, I accept this version of the manuscript.